environmental science/environmental engineering/computer modelling and simulation

urban heat island, green infrastructure, mitigation, spatial planning

**Author for correspondence:**
Martí Bosch
e-mail: marti.bosch@epfl.ch

# Evaluating urban greening scenarios for urban heat mitigation: a spatially explicit approach

Martí Bosch[1], Maxence Locatelli[1], Perrine Hamel[2], Roy P. Remme[3,4], Rémi Jaligot[1], Jérôme Chenal[1] and Stéphane Joost[1,5]

[1]Urban and Regional Planning Community, École Polytechnique Fédérale de Lausanne, Lausanne, Switzerland
[2]Asian School of the Environment, Nanyang Technological University, Singapore
[3]Institute of Environmental Sciences, Leiden University, Leiden, The Netherlands
[4]Natural Capital Project, Stanford University, Stanford, CA, USA
[5]Laboratory of Geographic Information Systems, École Polytechnique Fédérale de Lausanne, Lausanne, Switzerland

MB, 0000-0001-8735-9144; RPR, 0000-0002-0799-2319; SJ, 0000-0002-1184-7501

Urban green infrastructure, especially trees, are widely regarded as one of the most effective ways to reduce urban temperatures in heatwaves and alleviate the adverse impacts of extreme heat events on human health and well-being. Nevertheless, urban planners and decision-makers are still lacking methods and tools to spatially evaluate the cooling effects of urban green spaces and exploit them to assess greening strategies at the urban agglomeration scale. This article introduces a novel spatially explicit approach to simulate urban greening scenarios by increasing the tree canopy cover in the existing urban fabric and evaluating their heat mitigation potential. The latter is achieved by applying the InVEST urban cooling model to the synthetic land use/land cover maps generated for the greening scenarios. A case study in the urban agglomeration of Lausanne, Switzerland, illustrates the development of tree canopy scenarios following distinct spatial distribution strategies. The spatial pattern of the tree canopy strongly influences the human exposure to the highest temperatures, and small increases in the abundance of tree canopy cover with the appropriate spatial configuration can have major impacts on human health and well-being. The proposed approach supports urban planning and the design of nature-based solutions to enhance climate resilience.

# 1. Introduction

Urbanization is a global phenomenon that increasingly concentrates the world's population in urban areas, with the latter expected to grow in both the number of dwellers and spatial extent over the next decades [1–3]. As a major force of landscape change, urbanization is characterized by the conversion of natural to artificial surfaces, which alters the energy and water exchanges as well as the movement of air. Such changes often result in the urban heat island (UHI) effect, a phenomenon by which urban temperatures are warmer than its rural surroundings [4–9]. The negative impacts of UHI have been widely documented and include increased energy and water consumption [10–12], reduced workplace productivity [13,14] and aggravation of health risks [15–17]. As urban areas grow and global temperatures rise, the UHI effect is expected to become more intense [18,19], which makes urban heat mitigation a major priority for urban planning and policy-making [20].

Increasing urban green space, especially the urban tree canopy, has been one of the most widely advocated strategies of urban heat mitigation. Nevertheless, the impacts of the urban tree canopy on air temperature show a complex spatial behaviour that remains poorly understood [9,21,22]. While the evidence of the cooling effects of urban green areas has been extensively reported [21–27], the relationship between their size and their cooling capacity is nonlinear [28–31], and little is known about how the overall spatial configuration of urban green spaces affects the heat mitigation at the urban agglomeration scale [32–36]. Therefore, the way in which cities can plan and optimize their green infrastructure to reduce heat stress is not yet sufficiently understood, largely because of the lack of fine-grained approaches to evaluate the cooling effects of the spatial pattern of the tree canopy at the urban agglomeration scale.

With the aim of addressing the above shortcomings, the present work introduces a novel spatially explicit method to evaluate the heat mitigation potential of altering the abundance and spatial configuration of the urban tree canopy cover in a case study of the urban agglomeration of Lausanne, Switzerland. The proposed method consists of two major parts. First, synthetic scenarios are generated by increasing the tree canopy cover in candidate locations where the existing urban fabric permits it. Then, the spatial distribution of air temperature of each synthetic scenario is estimated with the InVEST urban cooling model, which simulates urban heat mitigation based on three biophysical processes, namely shade, evapotranspiration and albedo. Such a model has been calibrated and validated in the same study area in previous work [37]. Finally, the simulated temperature map is coupled with a gridded population census in order to evaluate the human exposure to urban heat in the scenario. By applying such a procedure in the urban agglomeration of Lausanne, Switzerland, this study aims to map the heat mitigation potential that can be achieved starting from the existing urban fabric. With the aim of quantifying the effects of the abundance and spatial configuration of the tree canopy cover on urban heat mitigation, a set of synthetic scenarios are generated by increasing different proportions of tree canopy cover in distinct spatial configurations. Although other studies have applied spatially explicit models of heat mitigation to evaluate future urbanization scenarios [38–40], the key novelty of this article is that scenarios are not defined to reflect specific planning strategies, e.g. business as usual, intensification, greening and the like, but rather to meticulously explore the effects of the spatial pattern of the tree canopy on urban heat mitigation.

# 2. Material and methods

## 2.1. Study area

Lausanne is the fourth largest Swiss urban agglomeration with 420 757 inhabitants as of January 2019 [41]. The agglomeration is located at the Swiss Plateau and on the shore of the Lake Léman, and is characterized by a continental temperate climate with mean annual temperatures of 10.9°C and mean annual precipitation of 100 mm, with a dominating vegetation of mixed broadleaf forest. The spatial extent of the study has been selected following the recent application of the InVEST urban cooling model to Lausanne by Bosch et al. [37], and covers an area of 112.46 km$^2$.

In order to evaluate the human exposure to UHI, the population data for the study area has been extracted from the population and households statistics (STATPOP) [42] provided at a 100 m resolution by the Swiss Federal Statistical Office (SFSO) with the Python library swisslandstats-geopy [43].

## 2.2. Simulation with the InVEST urban cooling model

The spatial distribution of air temperatures is simulated with the InVEST urban cooling model (version 3.8.0) [44], which is based on the heat mitigation provided by shade, evapotranspiration and albedo. The

main inputs are a land use/land cover (LULC) raster map, a reference evapotranspiration raster and a biophysical table containing model information of each LULC class of the map. The LULC maps have been obtained by rasterizing the vector geometries of the official cadastral survey of the Canton of Vaud [45] as of August 2019 to a 10 m resolution. Such a dataset distinguishes 25 LULC classes which are relevant to the urban, rural and wild landscapes encountered in Switzerland. The reference evapotranspiration pixel values are estimated with the Hargreaves equation [46] based on the daily minimum, average and maximum air temperature values of the 1 km gridded inventory of the Federal Office of Meteorology and Climatology (MeteoSwiss) [47]. The biophysical table used in this study is shown in electronic supplementary material, table S1. A more thorough description of the model and the data inputs can be found in Bosch *et al.* [37].

The parameters of the model are set based on its calibration to the same study area in previous work [37], which reproduces the air temperature measurements of 11 monitoring stations (see electronic supplementary material, figure S1) with a coefficient of adjustment ($R^2$) of 0.90 and a mean absolute error of 0.96°C. Finally, the temperature values observed at these stations are used to set the values of the rural reference temperature ($T_{ref}$) and UHI magnitude ($UHI_{max}$) parameters. More precisely, $T_{ref}$ is set as the air temperature measurement at 21.00—the moment of maximal UHI intensity in Switzerland [48]—of the station showing the lowest temperature value, and $UHI_{max}$ is set as the difference between the 21.00 temperature measurement of the station showing the highest temperature value and $T_{ref}$. With the above definitions, a reference day for the simulations has been selected from the 2018–2019 period as the day showing the maximum UHI magnitude, i.e. $UHI_{max}$, while ensuring that $T_{ref} > 20$. Such a date corresponds to 27 July 2018, with $T_{ref} = 20.60$°C and $UHI_{max} = 7.48$°C.

## 2.3. Refining LULC classes based on tree cover and building density

A procedure to redefine the LULC classes from the cadastral survey has been designed to distinguish the LULC classes depending on their proportional cover of both trees and buildings. The reclassification is achieved by combining the 10 m raster LULC map with two 1 m binary raster masks, one for the tree canopy raster and another for the buildings. The 1 m binary tree canopy mask has been derived from the SWISSIMAGE orthomosaic [49], by means of the Python library DetecTree [50], which implements the methods proposed by Yang *et al.* [51]. The estimated classification accuracy of the tree canopy classification is 91.75%. On the other hand, the 1 m binary building mask has been obtained by rasterizing the buildings of the vector cadastral survey [45].

The reclassification procedure consists of three steps. Firstly, each 10 m pixel is coupled with the tree canopy and building masks in order to respectively compute its proportion of tree and building cover. Secondly, the set of 10 m pixels of each LULC class are grouped into a user-defined set of bins to form two histograms, one based on their proportion of tree cover and the other analogously for the building cover. Lastly, the two histograms are joined so that each LULC class is further refined into a set of classes. For example, if two bins were used for both the tree and building cover, the 'sidewalk' LULC code might be further refined into 'sidewalk with low tree/low building cover', 'sidewalk with low tree/high building cover', 'sidewalk with high tree/low building cover' and 'sidewalk with high tree/high building cover'.

In the present work, four equally spaced bins (i.e. distinguishing 0–25%, 25–50%, 50–75% and 75–100% intervals) have been used to reclassify each LULC class according to both the tree and building cover. Following the advice given by the directorate of resources and natural heritage in the Canton of Vaud (DGE-DIRNA), the threshold over which a pixel is considered to have a high tree canopy cover has been set to 75%, which corresponds to placing trees of a spheric crown with a 5 m radius spaced 10 m from one another so that they form a continuous canopy. Finally, in order to adapt the biophysical table of the InVEST urban cooling model to the reclassified LULC classes, the shade coefficients are computed as the midpoint of the bin interval of each level of tree cover (i.e. 0.125, 0.375, 0.625 and 0.875), whereas the albedo coefficients have been linearly interpolated based on the level of building cover (see electronic supplementary material, table S1).

## 2.4. Generation of urban greening scenarios

Starting from the refined LULC map, a set of urban greening scenarios are generated by altering the LULC classes of certain candidate pixels in a way that corresponds to reasonable transformations that could occur in urban areas. More precisely, pixels whose base LULC class corresponds to 'building', 'road, path', 'sidewalk', 'traffic island', 'other impervious' and 'garden' are changed to the LULC code

**Table 1.** Selected landscape metrics. A more thorough description can be found in the documentation of the software FRAGSTATS v4 [53].

| category | metric name | description |
| --- | --- | --- |
| composition | percentage of landscape (PLAND) | percentage of landscape, in terms of area, occupied by pixels with high tree canopy cover |
| configuration | mean patch area (AREA_MN) | average size (in hectares) of the patches formed by pixels with high tree canopy cover |
| | mean shape index (SHAPE_MN) | average shape index of the patches formed by pixels with high tree canopy cover |
| | edge density (ED) | sum of the lengths of all edge segments between pixels with high tree canopy cover and other pixels, per area unit (in m ha$^{-1}$) |

that has the same base class but with the highest tree cover, e.g. pixels of a post-refinement class 'sidewalk with low tree/low building cover' are be changed to 'sidewalk with high tree/low building cover'. In order to ensure that such an increase of the tree canopy cover is performed only where the existing urban fabric permits it, pixels might only be transformed when two conditions are met. First, the proportion of building cover in the candidate pixels must be under 25%, i.e. there is 75% of the pixel area which could be occupied by a tree crown. Secondly, pixels of the 'road, path' class might only be transformed when they are adjacent to a pixel of a different class, which prevents increasing the tree canopy cover in pixels that are in the middle of a road (e.g. a highway).

After mapping the candidate pixels where the tree canopy cover can be increased, scenarios are generated based on two key attributes: the extent of tree canopy conversion (expressed as a proportion of the total number of candidate pixels), and the selection of pixels to be converted. A set of scenarios is generated by transforming 12.5, 25, 37.5, 50, 62.5, 75 and 87.5% of the candidate pixels, respectively. For each of these canopy areas, three distinct selection approaches are used. The first consists in randomly sampling from the candidate pixels until the desired proportion of changed pixels is matched. In the second and third approaches, the candidate pixels are sampled according to the number of pixels with high tree canopy cover (i.e. greater than 75%) found in their Moore neighbourhood (i.e. the eight adjacent pixels). In the second approach, pixels with higher number of high tree canopy cover neighbours are transformed first, which intends to spatially cluster pixels of high tree canopy cover. The third approach intends to spatially scatter pixels of high tree canopy cover by prioritizing pixels with lower number of high tree canopy neighbours. Given that the three sampling approaches are stochastic, for each scenario configuration, i.e. each pair of proportion of transformed candidate pixels and sampling approach, the corresponding temperature maps will be computed by averaging a number of simulation runs. After observing little variability among the simulation results, the number of runs for each configuration has been set to 10. Lastly, the set of scenarios is completed with a configuration where 100% of the candidate pixels are transformed, which is independent of the sampling approach or scenario run since there exists a single deterministic way to transform all the candidate pixels. The final number of simulated scenarios is 211, i.e. 10 scenario runs for three different sampling approaches and seven proportions of transformed candidate pixels, plus a last scenario where all the pixels are transformed.

For each scenario, the spatial pattern of the tree canopy is quantified by means of a set of spatial metrics from landscape ecology [52,53], which are computed for the pixels whose post-refinement LULC class has a tree canopy cover over 75%. As explained above, adjacent pixels with a tree canopy cover over 75% can be considered as forming a continuous canopy. Based on similar studies that explore the relationship between the spatial pattern of tree canopy and UHIs [36,54,55], four spatial metrics have been chosen to quantify both the composition and configuration of the tree canopy, which are listed in table 1. The proportion of landscape (PLAND) of pixels with high tree canopy cover serves to quantify the composition aspects, while the configuration is quantified by means of the mean patch size (MPS), edge density (ED) and the mean shape index (MSI) of patches of high tree canopy cover. The four metrics have been computed with the Python library PyLandStats [56].

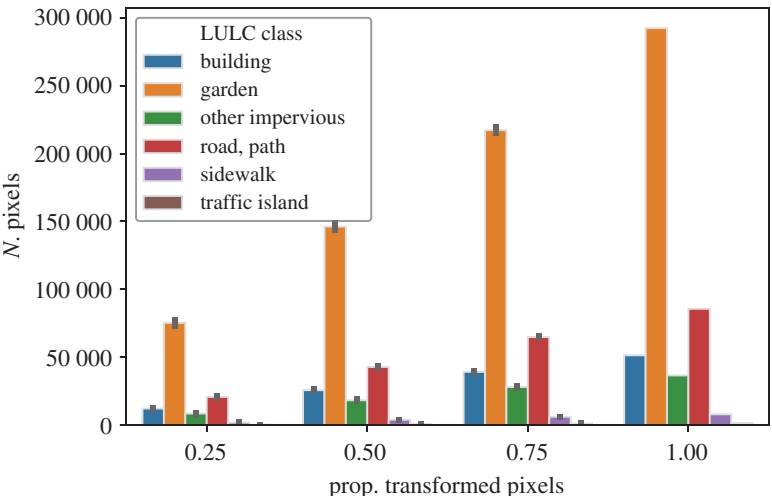

**Figure 1.** Number of transformed pixels by its original LULC class for an overall proportion of transformed pixels of 25, 50, 75 and 100%. The lines at the top of the bars represent the 95% confidence intervals. The bar heights and the confidence intervals are computed out of all the simulation runs and sampling approaches. See the Jupyter Notebook at electronic supplementary material, §S2.1 for the detailed numbers of the figure.

# 3. Results

## 3.1. Proportion of transformed pixels by their original LULC class

The relationship between the number of transformed candidate pixels by their original LULC class and the overall proportion of transformed candidate pixels is shown in figure 1. Changing 25, 50, 75 and 100% of the candidate pixels corresponds to a total number of pixels changed of 118 880, 237 760, 356 640 and 475 520, which account for a total area of 1188.8, 2377.6, 3566.4 and 4755.2 ha, respectively. In the last case, i.e. increasing the tree canopy in all the possible pixels, 61.50% of the pixels correspond to the 'garden' LULC class, followed by 'road, path', 'building', 'other impervious', (18.01, 10.81 and 7.69%, respectively). Finally, the LULC classes of 'sidewalk' and 'traffic island' constitute only 1.67 and 0.3% of the pixels where the tree canopy can be increased. The differences when considering the sampling approaches separately are small relative to the total number of transformed candidate pixels. The largest differences between sampling approaches can be noted in the number of transformed pixels that originally belong to the 'garden' class. When transforming 25, 50 and 75% of the candidate pixels, clustering, respectively, transforms (on average among the simulation runs) 0.90, 0.38 and 0.12% more garden pixels than random sampling, and 1.28, 0.76 and 0.43% more garden pixels than the scattering approach (figure 2).

## 3.2. Simulated LULC, temperature and heat mitigation maps

The LULC, temperature and heat mitigation maps for the scenarios generated by transforming 25, 50, 75 and 100% of the candidate pixels are shown in figure 3. When changing 25, 50, 75 and 100% of the candidate pixels, the maximum temperature $T$ for the reference date, i.e. 26.05°C, is progressively reduced to 25.77, 25.30, 24.82 and 24.49°C, respectively, while the magnitude of maximum heat mitigation $(T - T_{obs})$ increases from 0.49, 1.17, 1.81 and 2.22°C, respectively. The largest heat mitigation magnitudes occur in the most urbanized parts, which are located along the main transportation axes. The relationship between the proportion of candidate pixels transformed and the simulated distribution of air temperature can be approximated as a linear relationship with a negative slope (see electronic supplementary material, figures S2 and S3 for more details about this relationship).

## 3.3. Spatial patterns of tree canopy cover

The relationships between the landscape metrics of each scenario run and the corresponding simulated average temperature $\overline{T}$ (over all the pixels) are displayed in figure 4. The proportion of landscape (PLAND) occupied by pixels with high tree canopy cover range from 17.26 to 53.37%. As a

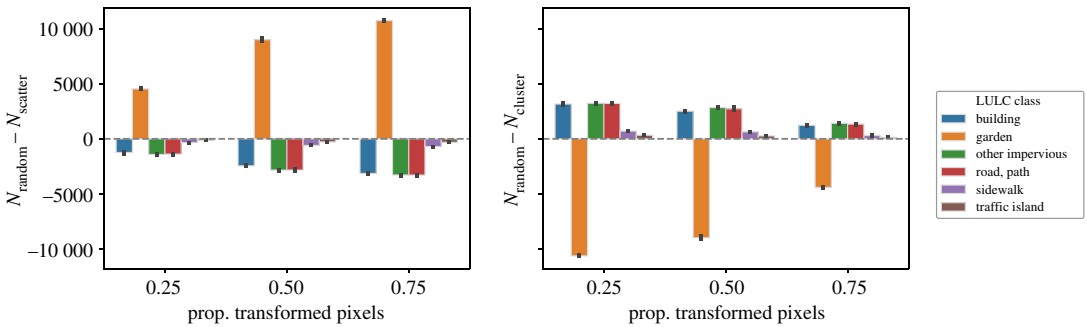

**Figure 2.** Comparison of between the number of transformed pixels by its original LULC class with the random sampling approach and the scattering ($N_{random} - N_{scatter}$, left subplot) and the clustering ($N_{random} - N_{cluster}$, right subplot) selection approaches, for an overall proportion of transformed pixels of 25, 50 and 75%. The lines at the top of the bars represent the 95% confidence intervals. The bar heights and the confidence intervals are computed out of all the simulation runs. See the Jupyter Notebook at electronic supplementary material, §S2.1 for the detailed numbers of the figure.

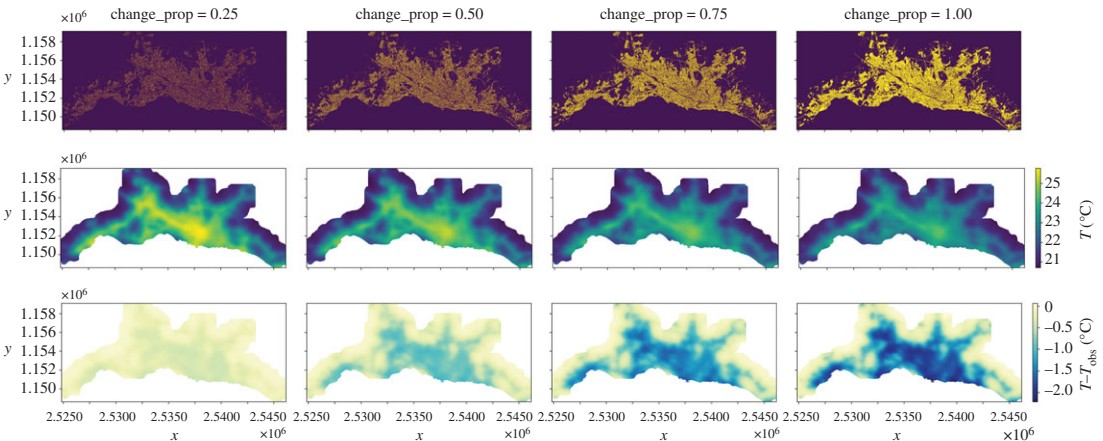

**Figure 3.** Simulated LULC (top, changed pixels are shown in yellow), temperature (middle) and heat mitigation (bottom) maps by transforming 25, 50, 75 and 100% of the candidate pixels to its corresponding LULC code with high tree canopy cover. The pixel values of each map are aggregated out of all the sampling approaches and simulation runs, i.e. the LULC maps show the mode, whereas the temperature and heat mitigation maps show the average. The axes tick labels display the Swiss CH1903+/LV95 coordinates. See the Jupyter Notebook at electronic supplementary material, §S2.1 for the detailed numbers of the figure.

composition metric, PLAND is directly related to the proportion of transformed candidate pixels, and the extreme values of the PLAND range correspond to transforming 0 and 100% of the candidate pixels respectively. The relationship between PLAND and the average simulated temperature of each scenario $\overline{T}$ shows a sharp monotonic decrease. However, for the same PLAND values, clustering the transformed pixels to other pixels with high tree canopy cover consistently leads to higher $\overline{T}$ than scattering or randomly sampling—the latter approaches show almost indistinguishable PLAND and $\overline{T}$ relationship.

Regarding the configuration metrics, the values of the mean patch area (AREA_MN) show that the clustering and random sampling approaches lead to larger patches of high tree canopy cover than the scattering approach. When transforming 12.5 and 25% of the candidate pixels, clustering them to other pixels of high tree canopy cover increases AREA_MN from 0.14 to 0.54 ha respectively (on average over the simulation runs). For 37.5% of transformed candidate pixels in the clustering approach, AREA_MN shows a sudden decline to 0.20 ha, followed by a monotonic increase that reaches 1.52 ha when all the candidate pixels are transformed. Such a discernable kink in the computed AREA_MN reveals characteristics of the existing urban fabric, and describes the point after which all the candidate pixels that are adjacent to other pixels of high tree canopy have been transformed and hence new pixels have to be allocated as part of new (and smaller) patches. The same kink is even more notable for the mean shape index (SHAPE_MN), yet the computed values show a very irregular pattern across the different scenario configurations, and it is the only metric

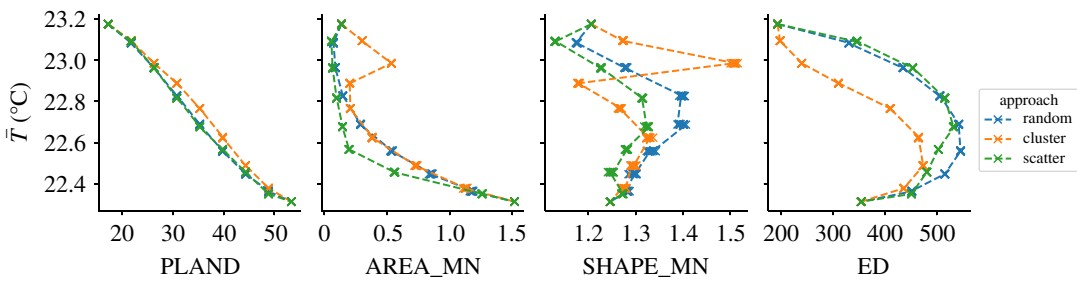

**Figure 4.** Relationship between landscape metrics and the simulated average temperature $\bar{T}$ for each scenario run, coloured to distinguish the sampling approaches. See the Jupyter Notebook at electronic supplementary material, §S2.2 for the detailed numbers of the figure.

where differences can be noted among scenario runs with the same configuration. The only consistency is that the scattering approach tends to lower SHAPE_MN values than randomly sampling the transformed pixels, which is probably due to the larger abundance of simple single-pixel patches in the former approach. Finally, the clustering approach results in lower edge density (ED) values than in the scattering and random sampling approaches, which show a very similar trend. The observed pattern is consistent with the notion that growing existing patches by clustering the new pixels to them accounts for less total edge length than scattering the same amount of new pixels in a leapfrog manner. In the three approaches, the ED increases monotonically at first until an apex is reached when the proportion of transformed pixels is between 50% and 60%, and then declines monotonically.

The average simulated temperature $\bar{T}$ is overall negatively correlated with AREA_MN, which suggests that for the same amount of high tree canopy pixels, large patches provide lower heat mitigation. On the other hand, configurations with the same proportion of high tree canopy pixels show lower $\bar{T}$ for larger values of ED, which suggests that edge effects between artificial patches and patches of high tree canopy contribute to greater heat mitigation. Nonetheless, as higher proportions of candidate pixels are transformed and the locations of the remaining candidate pixels force the overall ED to decrease, the simulated average temperatures continue to decline. This highlights how the cooling effects of the abundance of tree canopy overshadow those of the spatial configuration, which is consistent with many related studies [36,55,57–59].

## 3.4. Effects on human exposure

The relationship between human exposure to air temperatures higher than 21, 22, 23, 24, 25 and 26°C and the proportion of pixels transformed to their respective high tree canopy cover class is shown in figure 5. The number of dwellers exposed to temperatures higher than 21°C does not show a significant decrease (even when converting all the candidate pixels), whereas for temperatures higher than 22°C, it diminishes from 269 254 to 268 601, 267 683, 266 518 and 264 125 when the proportion of transformed pixels is 25, 50, 75 and 100%, respectively, which represents a relative share of 97.25, 97.02, 96.69 96.27 and 95.41% of the population of the study area. Such a decline progressively becomes more notable as temperatures increase, e.g. the share of the population exposed to temperatures over 24°C declines from an initial 78.4% to 72.39, 59.57, 37.53 and 11.52% when transforming 25, 50, 75 and 100% of the candidate pixels respectively. Finally, the share of dwellers exposed to temperatures over 25°C, which is initially 47.91%, is diminished to 24.98 and 5.74% when transforming 25 and 50% of the pixels, respectively, and becomes 0 after that, whereas the 2508 dwellers originally exposed to temperatures over 26°C do no longer meet such temperatures after transforming 25% of the candidate pixels.

The way in which the transformed pixels are sampled has significant effects on the human exposure to high temperatures (figure 6). Overall, scattering the transformed pixels to avoid forming a continuous tree canopy appears as the most effective approach to reduce the human exposure to the highest temperatures, followed by random sampling. When transforming 25 and 50% of the candidate pixels with the scattering approach, the number of dwellers exposed to temperatures over 25°C decreases from 124 073 to 65 108 and 4498 respectively. Such a reduction is larger than its random sampling counterpart by 3125 and 8223 dwellers, respectively, and larger than its clustering approach counterpart by 9359 and 21 388 dwellers, respectively (figure 6).

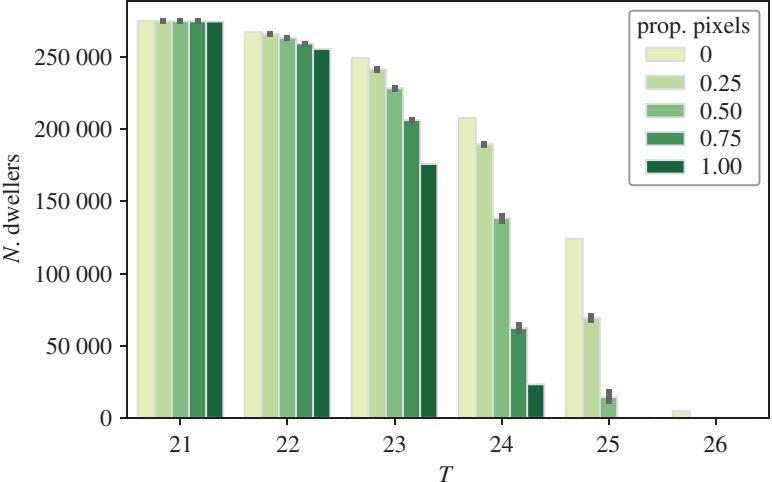

**Figure 5.** Population exposed to temperatures higher than 21, 22, 23, 24, 25 and 26°C respectively for an overall proportion of transformed pixels of 0, 25, 50, 75 and 100%. The bar heights and the confidence intervals are computed out of all the simulation runs and sampling approaches. See the Juptyer Notebook at electronic supplementary material, §S2.3 for the detailed numbers of the figure.

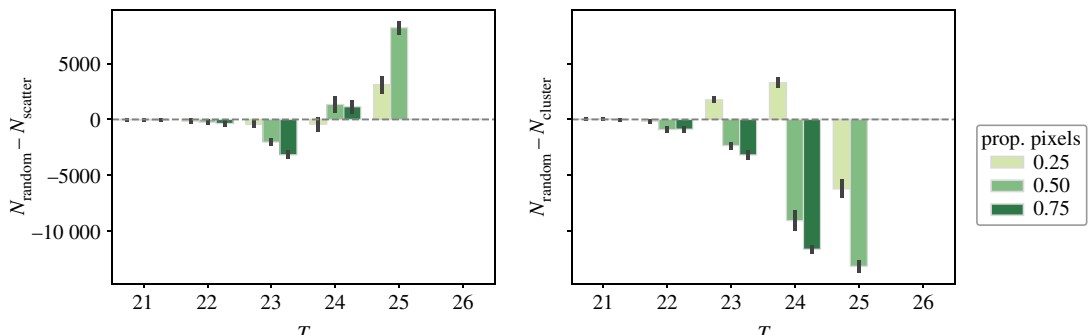

**Figure 6.** Comparison of the population exposed to temperatures higher than 21, 22, 23, 24, 25 and 26°C with the random sampling approach and the scattering ($N_\text{random} - N_\text{scatter}$, left subplot) and the clustering ($N_\text{random} - N_\text{cluster}$, right subplot) selection approaches, for an overall proportion of transformed pixels of 25, 50 and 75%. The lines at the top of the bars represent the 95% confidence intervals. The bar heights and the confidence intervals are computed out of all the simulation runs. See the Jupyter Notebook at electronic supplementary material, §S2.3 for the detailed numbers of the figure.

## 4. Discussion

### 4.1. Validity and applicability of the proposed approach

The scenarios simulated in this study map locations where the tree canopy cover in the urban agglomeration of Lausanne can be increased, and suggests that such changes can result in urban night-time temperatures that are up to 2°C lower. The results indicate that given the same proportion of tree canopy cover, a scattered configuration might lead to more effective urban heat mitigation than a clustered one, which is in line with previous studies in humid climates [54,55,60–63]. Nevertheless, the results suggest that the effect of the spatial configuration (measured by the metrics AREA_MN, SHAPE_MN and ED) is secondary when compared with the effect of the composition (measured by the PLAND metric). Overall, the effect of the spatial configuration of trees on its urban heat mitigation depends on how it affects the shading and evapotranspiration processes. Such a relationship is known to be strongly mediated by the tree species, background climatic and environmental conditions as well as the spatial scale [36,54,55,62,64–67].

The spatial effects observed in the results are due to the InVEST model equations representing air mixing and the effect of parks. In order to ascertain these effects, the InVEST urban cooling model must be further validated with experiments at the neighbourhood scale to ensure that it provides an

appropriate city-scale depiction of how the urban heat mitigation mechanisms operate at finer scales. In fact, the InVEST urban cooling model presents limitations regarding the simplified and homogeneous way in which the air is mixed, as well as the cooling effects of large green spaces [37,44]. As a result, the relationship between the proportion of tree canopy cover and the magnitude of the urban heat mitigation reported in this work is practically linear, and the temperature differences between spatially clustering or scattering the new tree canopy cover are limited. Nonetheless, in complex terrains such as the Lausanne agglomeration, models with uniform weighting of space show considerable deviations from the observed spatial patterns of air temperature [47,68]. Moreover, the cooling effects of large green spaces have been found to be non-proportional to their area and shape complexity [69–71]. Improving how these nonlinear components are represented in the InVEST urban cooling model could enhance not only its validity, but also its value to urban planning by identifying thresholds and regime changes in the cooling efficiency of additional tree planting. Another major limitation of the InVEST urban cooling model is that it only considers the shade cast by trees, hence overlooking the shade cast by buildings, which also has significant cooling effects [72–77]. Therefore, in order to improve the ability of the model to accurately represent physical processes associated with the heat mitigation, the model should be extended to include a more detailed representation of the three-dimensional features of the urban canyon.

Despite the limitations noted above, a major advantage of the proposed approach is that it can be used to evaluate urban heat mitigation of synthetic scenarios. The simulations presented in this article focus on spatially exploring the effects of an increase of the tree canopy cover, yet there is room for much more experimentation of this kind. On the one hand, the generic sampling approaches explored above can be extended to consider ad hoc characteristics such as the spatial distribution of the population, and design optimization procedures with specific goals. For instance, the candidate pixels can be selected with the aim of minimizing the exposure of the most vulnerable populations to critical heat thresholds. More broadly, the approach can be used as part of a decision support system to explore the trade-offs between ecosystem services provided by trees, perform weighted optimizations and map priority planting locations [78,79]. On the other hand, in line with recent studies [80–82], the approach could be applied to examine the impact of distinct urbanization scenarios such as densification and urban sprawl on air temperature and human exposure to extreme heat, under current conditions as well as future climate estimates, e.g. by changing the $T_{ref}$ or $UHI_{max}$ parameters. Similarly, the InVEST urban cooling model might be coupled with models of LULC change such as cellular automata in order to assess not only which scenarios are most desirable in terms of urban heat mitigation, but also which planning strategies might lead to them [83–85].

## 4.2. Implications for urban planning in Lausanne

The spatio-temporal patterns of LULC change observed during the last 40 years in the Lausanne agglomeration have been characterized by infilling development and a progressive coalescence of artificial surfaces in its inner ring [86]. Such an infilling trend urges for careful evaluation of the beneficial ecosystem services provided by urban green spaces, which should be balanced against the adverse consequences of urban sprawl [34,35].

The approach proposed in this study maps locations in the current urban fabric where the tree canopy cover can be increased. While part of this urban greening might occur in impervious surfaces (e.g. in sidewalks, next to roads and in other impervious surfaces), most of the candidate locations currently correspond to urban green space (i.e. the 'garden' LULC class). Therefore, the potential heat mitigation suggested by the results of the study is not attainable in a scenario of severe infill development. Additionally, densification strategies should consider that newly created urban green space might result in less provision of ecosystem services than remnant natural patches [33,87,88]. Finally, infilling might exacerbate the unevenness of the accessibility to green areas by depriving dwellers of the most dense parts in city core from their few remaining urban green spaces. Spatial heterogeneity of this kind, which is encountered in many socioeconomic and environmental aspects of contemporary cities, is often hard to represent with aggregate indicators and highlights the importance of spatially explicit models to urban planning and decision making.

The explicit representation of space is also crucial when considering the impacts of urban green space on human exposure to extreme heat. Although the simulated scenarios suggest that the impact of the spatial pattern of tree canopy on the air temperature is practically linear, the implications on human exposure to critical temperatures exhibit important thresholds. For example, by increasing the tree canopy cover of 25% of the candidate pixels, the number of dwellers exposed to night-time

temperatures over 25°C can be reduced from 124 073 to 74 466, which, respectively, represents 45.08 and 27.06% of the total population in the study area. Furthermore, the results suggest by selecting such pixels to prioritize a spatial scattering of the tree canopy cover, such a population can be reduced by an additional 3125 or 6234 dwellers when, respectively, compared with random sampling such pixels or clustering them to the existing tree canopy cover. In Switzerland, the excess mortality associated to the heat wave of 2003 occurred disproportionately to urban and sub-urban residents of its largest urban agglomerations [89]. Furthermore, the association between temperature and mortality in extreme heat events in the largest Swiss urban agglomerations are exponential [90], which indicates that reducing temperatures by even fractions of a degree can have a dramatic impact on death rates.

## 5. Conclusion

The scenarios simulated in this study represent a new way of spatially exploring the heat mitigation potential provided by modifications of the urban fabric, and allow evaluating the cooling effects of both the abundance and spatial configuration of the tree canopy cover. The results map locations where the existing tree canopy cover of the urban agglomeration of Lausanne can be increased, and show an urban cooling potential for urban night-time temperatures of more than 2°C. Additionally, the simulations suggest that the spatial configuration in which the tree canopy is increased influences its heat mitigation effects. The configuration effects become more significant when considering the impacts on the urban population, and small increases in the tree canopy can result in important reductions in the number of dwellers exposed to the highest temperatures. Overall, the presented approach provides a novel way to explore how the urban tree canopy can be exploited to reduce heat stress. Future studies can extend the analyses by assessing the provision of other ecosystem services in the various tree canopy strategies presented here.

Data accessibility. The data required to reproduce the results is available at https://zenodo.org/record/4316572. A repository with the code materials and detailed instructions of the steps to reproduce the results is available at https://github.com/martibosch/lausanne-greening-scenarios.
Competing interests. We declare we have no competing interests.
Funding. This research has been supported by the École Polytechnique Fédérale de Lausanne (EPFL). The authors received no specific funding for this work.

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
