## [Peer Review File · Royal Society Open Science]

Review History

RSOS-202174.R0 (Original submission)

Review form: Reviewer 1 (Brenda Lin)

Is the manuscript scientifically sound in its present form?

Yes

Are the interpretations and conclusions justified by the results?

Yes

Is the language acceptable?

Yes

Do you have any ethical concerns with this paper?

No

Have you any concerns about statistical analyses in this paper?

No

Recommendation?

Accept with minor revision (please list in comments)

Comments to the Author(s)

I found this paper to be interesting in its approach. The authors are correct in stating that many governments are uncertain about how to measure the impact of different spatial configurations of trees on air temperature. Having a robust modelling approach will be helpful in moving this difficult area forward for decision-makers.

There are a couple of small comments that I think would make the manuscript more robust.

Abstract:

- “and alleviate its adverse impacts on human health and well-being.”

A bit awkward – its seems to refer to trees, not extreme heat events

Intro

- L31 – “Then, the spatial distribution of air temperature of each 30 synthetic scenario is estimated with the InVEST urban cooling model, which simulates 31 urban heat mitigation based on three biophysical processes, namely shade, 32 evapotranspiration and albedo.”

How well does the InVEST model measure and model urban cooling though? This needs to be supported with references that have validated the model as a good model to use to understand if it can accurately estimate changes in air temperature with changes in spatial configuration of trees.

I see that this information is in the methods, but in the introduction, it would be good to cite some of the work that has already shown that this is a validated model.

- Perhaps a framework figure would be good at the end of the introduction.

Methods:

- L78 - Is the reference day chosen because it was an average day for the summer? If not, please explain how it was chosen. It does not seem to be very warm for a summer day? More information on why this was chosen needs to be provided to support the analysis.

- One question I have with the classification is that there is so much variability in land use types and how they impact UHI. For example, a building of two different materials will have a different impact on the local microclimate. The same could be said for a road or path that is built with, for example, a pervious pavement. Gardens are also highly variable in their management and how they impact UHI. How do we standardise this in a model while maintaining some level of reality? I suppose we have to assume that the comparisons are against the same standardisation, and that provides us a sense of how spatial configurations can be better, although there can be variation in the level of impact.

Discussion:

- I have another question around the temperatures thresholds selected. I think at 21-26 degrees, is there evidence that these temperatures badly affect the residents in these areas? As someone who probably mainly is exposed to temperature above 25 degrees, I am probably more accustomed to warmer temperatures. There probably needs to be some discussion around human health and comfort and how this changes through exposure. Maybe some literature on what the thresholds are in the literature for this area.

- I'm glad that there is a good discussion on the limitations of the model. I think it's important that these points are made. I think this provides room for the authors to discuss what contribution the model is making though – e.g. because the model provides an approximation of comparison...

Review form: Reviewer 2

Is the manuscript scientifically sound in its present form?

No

Are the interpretations and conclusions justified by the results?

Yes

Is the language acceptable?

Yes

Do you have any ethical concerns with this paper?

No

Have you any concerns about statistical analyses in this paper?

No

Recommendation?

Major revision is needed (please make suggestions in comments)

Comments to the Author(s)

This paper presents a very interesting piece of research that demonstrates a spatially-explicit simulation method for evaluating the heat mitigation potential of modifying the abundance and configuration of urban tree canopy cover, with a case study of an urban agglomeration in Swiss. Overall, the paper is well written in academic English with a clear description of the methodology and materials. The topic, how many trees to plant and where, is an important, growing research field, which is examined in this paper with advanced software and programming techniques. As noted in the paper, the proposed method has high potential to be incorporated with cellular automata for an integrated analysis of LULC changes and heat risk. While the substances in this paper are believed to merit publication, there are several questionable methodological decisions made by the authors.

1. Lines 92-101: regarding the reclassification, is it correct that high tree and high building covers are compatible for any LULC class? How are a 75% coverage of trees and another 75% coverage of building pixels allowed at the same time?

2. Lines 110-112: Is it possible that the InVEST urban cooling model considers the shades cast by buildings as well as by trees? If not, how does your model factor in building shade effects in the microclimate simulation? The shades created by buildings have as significant cooling effects on nearby surfaces as tree canopies. See recent empirical findings on building shade:

Park, Y., Guldmann, J. M., & Liu, D. (2021). Impacts of tree and building shades on the urban heat island: Combining remote sensing, 3D digital city and spatial regression approaches. *Computers, Environment and Urban Systems*, 88, pp.101655.

Hu, Y., Dai, Z., & Guldmann, J. M. (2020). Modeling the impact of 2D/3D urban indicators on the urban heat island over different seasons: A boosted regression tree approach. *Journal of environmental management*, 266, pp.110424.

3. Lines 154-156: Would there be any possibility of misrepresentation of the spatial pattern of tree canopies if using only those pixels with tree canopy cover >75% to compute landscape metrics? A

tree canopy cover between 50% and 75% is not negligible at all, possibly generating a considerable amount of shades and evapotranspiration. I understand that, in Fragstats, landscape metrics are computable only for discretely-coded pixels (tree vs. non-tree), not for continuous-valued pixels (e.g., percentage of tree cover). There is a recent study that develops a method to compute landscape patterns based on continuous pixel values, with an application to tree canopy percent maps:

Park, Y., & Guldman, J-M. (2020). Measuring continuous landscape patterns with Gray-Level Co-Occurrence Matrix (GLCM) indices: An alternative to patch metrics?. *Ecological Indicators*, 109, pp.105802.

4. Is it correct that pixels whose tree cover % is greater than 75 are only selected to compute PLAND (Table 1)? Why not computing PLAND just based on the continuous tree cover raster?

5. Another comment on the selection of landscape metrics as shown in Table 1. The landscape metrics under the category of configuration in Table 1 are about area & edge (AREA_MN, ED) and shape complexity (SHAPE_MN). There are other sub-categories of configuration (McGarigal 2015). Clarify the criteria taken when selecting this list of metrics while excluding other configuration metrics (e.g., aggregation, core area).

6. Figure 4: These figures are difficult to understand, particularly SHAPE_MN and ED. How could it be explained that the same values of ED (or SHAPE_MN) can lead to quite different air temperatures?

Lastly, there are quite a number of typos throughout the paper. A careful proofread is needed.

For example:

Line 62: ot -> to

Line 65: of by -> by

Lines 108-109: 'therefore, adjacent pixels with a tree canopy cover over 75% can'. -> ???

Line 147: of a -> for

Line 151: scenarios simulated scenarios -> simulated scenarios

Footnote 1: this is a repeat of lines 104-108.

Line 157: the spatial of tree canopy -> the spatial distribution of tree canopy?

Line 351: over-proportionally -> disproportionately

Decision letter (RSOS-202174.R0)

Dear Dr Bosch,

The Editors assigned to your paper RSOS-202174 "Evaluating urban greening scenarios for urban heat mitigation: a spatially-explicit approach" have now received comments from reviewers and would like you to revise the paper in accordance with the reviewer comments and any comments from the Editors. Please note this decision does not guarantee eventual acceptance.

Firstly, please accept our sincere apologies for the unusual delays incurred during the review process. We regret that it took far longer than usual to acquire reviewers, and owing to staff and Editor absences related to the pandemic, this also caused some delays to the process. We will endeavour to do all that we can to expedite your paper once you have submitted a revised version. We invite you to respond to the comments supplied below and revise your manuscript. Below the referees' and Editors' comments (where applicable) we provide additional requirements. Final acceptance of your manuscript is dependent on these requirements being met. We provide guidance below to help you prepare your revision.

Note that the decision that significant revision is required is based on the fact that whilst both reviewers make positive comments about the paper, Reviewer 2 in particular provides a comprehensive and sensible set of review comments that require consideration and response before the paper can proceed to publication.

Please submit your revised manuscript and required files (see below) no later than 21 days from today's (ie 08-Jul-2021) date. Note: the ScholarOne system will 'lock' if submission of the revision is attempted 21 or more days after the deadline. If you do not think you will be able to meet this deadline please contact the editorial office immediately.

on behalf of Dr Yhasmin Mendes de Moura (Associate Editor) and Peter Haynes (Subject Editor)
openscience@royalsociety.org

Reviewer comments to Author:

Reviewer: 1

Comments to the Author(s)

I found this paper to be interesting in its approach. The authors are correct in stating that many governments are uncertain about how to measure the impact of different spatial configurations of trees on air temperature. Having a robust modelling approach will be helpful in moving this difficult area forward for decision-makers.

There are a couple of small comments that I think would make the manuscript more robust.

Abstract:

- "and alleviate its adverse impacts on human health and well-being."

A bit awkward - its seems to refer to trees, not extreme heat events

Intro

- L31 - "Then, the spatial distribution of air temperature of each 30 synthetic scenario is estimated with the INVEST urban cooling model, which simulates 31 urban heat mitigation based on three biophysical processes, namely shade, 32 evapotranspiration and albedo."

How well does the InVEST model measure and model urban cooling though? This needs to be supported with references that have validated the model as a good model to use to understand if it can accurately estimate changes in air temperature with changes in spatial configuration of trees.

I see that this information is in the methods, but in the introduction, it would be good to cite some of the work that has already shown that this is a validated model.

- Perhaps a framework figure would be good at the end of the introduction.

Methods:

- L78 - Is the reference day chosen because it was an average day for the summer? If not, please explain how it was chosen. It does not seem to be very warm for a summer day? More information on why this was chosen needs to be provided to support the analysis.

- One question I have with the classification is that there is so much variability in land use types and how they impact UHI. For example, a building of two different materials will have a different impact on the local microclimate. The same could be said for a road or path that is built with, for example, a pervious pavement. Gardens are also highly variable in their management and how they impact UHI. How do we standardise this in a model while maintaining some level of reality? I suppose we have to assume that the comparisons are against the same standardisation, and that provides us a sense of how spatial configurations can be better, although there can be variation in the level of impact.

Discussion:

- I have another question around the temperatures thresholds selected. I think at 21-26 degrees, is there evidence that these temperatures badly affect the residents in these areas? As someone who probably mainly is exposed to temperature above 25 degrees, I am probably more accustomed to warmer temperatures. There probably needs to be some discussion around human health and comfort and how this changes through exposure. Maybe some literature on what the thresholds are in the literature for this area.

- I'm glad that there is a good discussion on the limitations of the model. I think it's important that these points are made. I think this provides room for the authors to discuss what contribution the model is making though - e.g. because the model provides an approximation of comparison...

Reviewer: 2

Comments to the Author(s)

This paper presents a very interesting piece of research that demonstrates a spatially-explicit simulation method for evaluating the heat mitigation potential of modifying the abundance and configuration of urban tree canopy cover, with a case study of an urban agglomeration in Swiss. Overall, the paper is well written in academic English with a clear description of the methodology and materials. The topic, how many trees to plant and where, is an important, growing research field, which is examined in this paper with advanced software and programming techniques. As noted in the paper, the proposed method has high potential to be incorporated with cellular automata for an integrated analysis of LULC changes and heat risk. While the substances in this paper are believed to merit publication, there are several questionable methodological decisions made by the authors.

1. Lines 92-101: regarding the reclassification, is it correct that high tree and high building covers are compatible for any LULC class? How are a 75% coverage of trees and another 75% coverage of building pixels allowed at the same time?

2. Lines 110-112: Is it possible that the InVEST urban cooling model considers the shades cast by buildings as well as by trees? If not, how does your model factor in building shade effects in the microclimate simulation? The shades created by buildings have as significant cooling effects on nearby surfaces as tree canopies. See recent empirical findings on building shade:
 Park, Y., Guldmann, J. M., & Liu, D. (2021). Impacts of tree and building shades on the urban heat island: Combining remote sensing, 3D digital city and spatial regression approaches. *Computers, Environment and Urban Systems*, 88, pp.101655.
 Hu, Y., Dai, Z., & Guldmann, J. M. (2020). Modeling the impact of 2D/3D urban indicators on the urban heat island over different seasons: A boosted regression tree approach. *Journal of environmental management*, 266, pp.110424.
3. Lines 154-156: Would there be any possibility of misrepresentation of the spatial pattern of tree canopies if using only those pixels with tree canopy cover >75% to compute landscape metrics? A tree canopy cover between 50% and 75% is not negligible at all, possibly generating a considerable amount of shades and evapotranspiration. I understand that, in Fragstats, landscape metrics are computable only for discretely-coded pixels (tree vs. non-tree), not for continuous-valued pixels (e.g., percentage of tree cover). There is a recent study that develops a method to compute landscape patterns based on continuous pixel values, with an application to tree canopy percent maps:
 Park, Y., & Guldmann, J-M. (2020). Measuring continuous landscape patterns with Gray-Level Co-Occurrence Matrix (GLCM) indices: An alternative to patch metrics?. *Ecological Indicators*, 109, pp.105802.
4. Is it correct that pixels whose tree cover % is greater than 75 are only selected to compute PLAND (Table 1)? Why not computing PLAND just based on the continuous tree cover raster?
5. Another comment on the selection of landscape metrics as shown in Table 1. The landscape metrics under the category of configuration in Table 1 are about area & edge (AREA_MN, ED) and shape complexity (SHAPE_MN). There are other sub-categories of configuration (McGarigal 2015). Clarify the criteria taken when selecting this list of metrics while excluding other configuration metrics (e.g., aggregation, core area).
6. Figure 4: These figures are difficult to understand, particularly SHAPE_MN and ED. How could it be explained that the same values of ED (or SHAPE_MN) can lead to quite different air temperatures?
- Lastly, there are quite a number of typos throughout the paper. A careful proofread is needed. For example:
 Line 62: ot -> to
 Line 65: of by -> by
 Lines 108-109: 'therefore, adjacent pixels with a tree canopy cover over 75% can'. -> ???
 Line 147: of a -> for
 Line 151: scenarios simulated scenarios -> simulated scenarios
 Footnote 1: this is a repeat of lines 104-108.
 Line 157: the spatial of tree canopy -> the spatial distribution of tree canopy?
 Line 351: over-proportionally -> disproportionately

===PREPARING YOUR MANUSCRIPT===

===PREPARING YOUR REVISION IN SCHOLARONE===

<https://royalsociety.org/journals/authors/author-guidelines/#supplementary-material> to include a suitable title and informative caption. An example of appropriate titling and captioning may be found at https://figshare.com/articles/Table_S2_from_Is_there_a_trade-off_between_peak_performance_and_performance_breadth_across_temperatures_for_aerobic_sc_ope_in_teleost_fishes_/3843624.

Author's Response to Decision Letter for (RSOS-202174.R0)

See Appendix A.

Decision letter (RSOS-202174.R1)

Dear Dr Bosch,

It is a pleasure to accept your manuscript entitled "Evaluating urban greening scenarios for urban heat mitigation: a spatially-explicit approach" in its current form for publication in Royal Society Open Science.

Please ensure that you send to the editorial office an editable version of your accepted manuscript, and individual files for each figure and table included in your manuscript. You can send these in a zip folder if more convenient. Failure to provide these files may delay the processing of your proof.

on behalf of Dr Yhasmin Mendes de Moura (Associate Editor) and Professor Peter Haynes (Subject Editor)
openscience@royalsociety.org

Appendix A

Reviewer comments to Author

Reviewer: 1

Comments to the Author(s)

I found this paper to be interesting in its approach. The authors are correct in stating that many governments are uncertain about how to measure the impact of different spatial configurations of trees on air temperature. Having a robust modelling approach will be helpful in moving this difficult area forward for decision-makers.

There are a couple of small comments that I think would make the manuscript more robust.

Abstract: - “and alleviate its adverse impacts on human health and well-being.”

A bit awkward – its seems to refer to trees, not extreme heat events

The sentence has been changed to “Urban green infrastructure, especially trees, are widely regarded as one of the most effective ways to reduce urban temperatures in heatwaves, and alleviate *the adverse impacts of extreme heat events* on human health and well-being.”

Intro - L31 – “Then, the spatial distribution of air temperature of each 30 synthetic scenario is estimated with the InVEST urban cooling model, which simulates 31 urban heat mitigation based on three biophysical processes, namely shade, 32 evapotranspiration and albedo.”

How well does the InVEST model measure and model urban cooling though? This needs to be supported with references that have validated the model as a good model to use to understand if it can accurately estimate changes in air temperature with changes in spatial configuration of trees.

I see that this information is in the methods, but in the introduction, it would be good to cite some of the work that has already shown that this is a validated model.

A new sentence “Such a model has been calibrated and validated in the same study area in previous work [37]” (reference 37 points to our previous work on the calibration and validation of the model in the same study area) has been added in lines 33-34.

- Perhaps a framework figure would be good at the end of the introduction.

Methods: - L78 - Is the reference day chosen because it was an average day for the summer? If not, please explain how it was chosen. It

does not seem to be very warm for a summer day? More information on why this was chosen needs to be provided to support the analysis.

Since we intend to study the UHI effect the reference day is chosen as the day with largest UHI magnitude while ensuring that it is a “hot” day (which is why we ensure that the minimum temperatures are over 20 C. It may not seem very hot but keep in mind that we are talking about 9pm temperatures and that Lausanne has relatively cool summers. The reference day is actually among the 15% hottest days of the summers of 2018 and 2019 (I could also compute such a percentile over the whole year). In any case, we have made some small changes in lines 86-87 which we hope that make the choice of a reference day clearer.

- One question I have with the classification is that there is so much variability in land use types and how they impact UHI. For example, a building of two different materials will have a different impact on the local microclimate. The same could be said for a road or path that is built with, for example, a pervious pavement. Gardens are also highly variable in their management and how they impact UHI. How do we standardise this in a model while maintaining some level of reality? I suppose we have to assume that the comparisons are against the same standardisation, and that provides us a sense of how spatial configurations can be better, although there can be variation in the level of impact.

Our reclassification based on building and tree cover tries to address this issue and is reflected in the impact on the UHI effect since we adjust the shade and albedo weights for the InVEST urban cooling model accordingly. Another (more standardized) approach would be using local climate zones, which we would like to link with the InVEST urban cooling model in future research.

Discussion: - I have another question around the temperatures thresholds selected. I think at 21-26 degrees, is there evidence that these temperatures badly affect the residents in these areas? As someone who probably mainly is exposed to temperature above 25 degrees, I am probably more accustomed to warmer temperatures. There probably needs to be some discussion around human health and comfort and how this changes through exposure. Maybe some literature on what the thresholds are in the literature for this area.

Like in the choice of the reference days, the temperature thresholds may not seem too high in many regions in the world but they are quite high for 9 p.m. temperatures in Lausanne.

- I’m glad that there is a good discussion on the limitations of the model. I think it’s important that these points are made. I think this provides room for the authors to discuss what contribution the model is making though – e.g. because the model provides an approximation of comparison...

We have tried to be very thorough in the discussion of the limitations because it is one of the first applications of the InVEST urban cooling model and we believe that there are many research directions that have to be explored - we try to summarize the most important ones in the discussion.

Reviewer: 2

Comments to the Author(s)

This paper presents a very interesting piece of research that demonstrates a spatially-explicit simulation method for evaluating the heat mitigation potential of modifying the abundance and configuration of urban tree canopy cover, with a case study of an urban agglomeration in Swiss. Overall, the paper is well written in academic English with a clear description of the methodology and materials. The topic, how many trees to plant and where, is an important, growing research field, which is examined in this paper with advanced software and programming techniques. As noted in the paper, the proposed method has high potential to be incorporated with cellular automata for an integrated analysis of LULC changes and heat risk. While the substances in this paper are believed to merit publication, there are several questionable methodological decisions made by the authors.

1. Lines 92-101: regarding the reclassification, is it correct that high tree and high building covers are compatible for any LULC class? How are a 75% coverage of trees and another 75% coverage of building pixels allowed at the same time?

As explained in lines 129-136, the procedure does exclude LULC classes where tree and building cover add up to more than 100%.

2. Lines 110-112: Is it possible that the InVEST urban cooling model considers the shades cast by buildings as well as by trees? If not, how does your model factor in building shade effects in the microclimate simulation? The shades created by buildings have as significant cooling effects on nearby surfaces as tree canopies. See recent empirical findings on building shade: Park, Y., Guldmann, J. M., & Liu, D. (2021). Impacts of tree and building shades on the urban heat island: Combining remote sensing, 3D digital city and spatial regression approaches. *Computers, Environment and Urban Systems*, 88, pp.101655. Hu, Y., Dai, Z., & Guldmann, J. M. (2020). Modeling the impact of 2D/3D urban indicators on the urban heat island over different seasons: A boosted regression tree approach. *Journal of environmental management*, 266, pp.110424.

We have added a paragraph in the discussion (lines 306-311) to acknowledge the major limitation of omitting shades casted by buildings. We are actually working in another article to further validate the model with a dense network

of temperature sensors, where we plan to include three-dimensional features of the urban canyon (such as the volume and space between buildings, street orientations and the like). However, we barely just collected the data and the article will still take two or three months.

3. Lines 154-156: Would there be any possibility of misrepresentation of the spatial pattern of tree canopies if using only those pixels with tree canopy cover >75% to compute landscape metrics? A tree canopy cover between 50% and 75% is not negligible at all, possibly generating a considerable amount of shades and evapotranspiration. I understand that, in Fragstats, landscape metrics are computable only for discretely-coded pixels (tree vs. non-tree), not for continuous-valued pixels (e.g., percentage of tree cover). There is a recent study that develops a method to compute landscape patterns based on continuous pixel values, with an application to tree canopy percent maps: Park, Y., & Guldmann, J-M. (2020). Measuring continuous landscape patterns with Gray-Level Co-Occurrence Matrix (GLCM) indices: An alternative to patch metrics?. *Ecological Indicators*, 109, pp.105802.
4. Is it correct that pixels whose tree cover % is greater than 75 are only selected to compute PLAND (Table 1)? Why not computing PLAND just based on the continuous tree cover raster?

The whole idea behind using pixels with more than 75 percent tree cover is that it eases the simulation of future scenarios: with the pixel width of 10 m, planting a tree can be represented by increasing the tree cover of a given pixel by 50 or 75 percent. Simulating tree planting based on the continuous tree cover raster would be much more complicated. We are aware that ommitting pixels with some canopy cover which can have important cooling effects in the end. However, at this stage (kind of proof-of context) our approach does not focus so much on the precision but rather on exploring hundreds of scenarios to get fine grained insights into the effects of the spatial pattern on tree canopy.

5. Another comment on the selection of landscape metrics as shown in Table 1. The landscape metrics under the category of configuration in Table 1 are about area & edge (AREA_MN, ED) and shape complexity (SHAPE_MN). There are other sub-categories of configuration (McGarigal 2015). Clarify the criteria taken when selecting this list of metrics while excluding other configuration metrics (e.g., aggregation, core area).

We have added references to justify the choice of landscape metrics. LPI has been excluded because the agglomeration extent “trims” the largest forest patches in the outskirts so the metric would not make much sense.

6. Figure 4: These figures are difficult to understand, particularly

SHAPE_MN and ED. How could it be explained that the same values of ED (or SHAPE_MN) can lead to quite different air temperatures?

The same values of edge density or shape index can lead to quite different air temperatures because they have different amounts of tree canopy cover (reflected by the proportion of landscape metric). This is consistent with what we discuss in line 251 “the cooling effects of the abundance of tree canopy overshadow those of the spatial configuration’

Lastly, there are quite a number of typos throughout the paper. A careful proofread is needed. For example: Line 62: ot -> to Line 65: of by -> by Lines 108-109: ‘therefore, adjacent pixels with a tree canopy cover over 75% can’. -> ??? Line 147: of a -> for Line 151: scenarios simulated scenarios -> simulated scenarios Footnote 1: this is a repeat of lines 104-108. Line 157: the spatial of tree canopy -> the spatial distribution of tree canopy? Line 351: over-proportionally -> disproportionately

We believe that we have corrected all these (and other) typos. We thank the reviewer for pointing them out.